# A Wideband Low-Power Balun-LNA with Feedback and Current Reuse Technique

Muhammad Fakhri Mauludin [1], Dong-Ho Lee [2] and Jusung Kim [1,*]

[1] Department of Electronics Engineering, Hanbat National University, Daejeon 34158, Korea; 30204004@o365.hanbat.ac.kr
[2] Department of Mobile Convergence Engineering, Hanbat National University, Daejeon 34158, Korea; dhlee@hanbat.ac.kr
* Correspondence: jusungkim@hanbat.ac.kr

**Abstract:** This paper presents a low-noise amplifier (LNA) with single to differential conversion (Balun) for multi-standard radio applications. The proposed LNA combines a common-gate (CG) stage for wideband input matching and a common-source (CS) stage to cancel the noise and distortion of the CG stage. Using the proposed technique, a low noise figure (NF) is achieved while providing a wideband of operation. Furthermore, a feedback connection from the CS stage to the gate of the CG is employed to boost the transconductance of the CG stage ($gm_{CG}$), and an additional complementary transistor is applied at the CS stage using current reuse to increase the overall transconductance of the CS stage ($gm_{CS}$) without increasing the power consumed by the stage. This LNA was designed using TSMC 65 nm technology, and post-layout simulation results show operation across 0.5–5 GHz, a maximum power gain of 20 dB, 4 dB minimum NF, and third-order intercept point (IIP3) of −10 dBm while consuming only 5 mW of power from a 1.2 V supply.

**Keywords:** balun; CMOS; common-gate (CG) amplifier; common-source (CS) amplifier; low-noise amplifier (LNA); noise cancelation; noise reduction; wideband





## 1. Introduction

Wireless communication systems have evolved rapidly and are now among the most important aspects of daily life. Various wireless standards, such as Bluetooth, Wi-Fi, GPS, and 2G/3G/4G/5G cellular systems, use different frequency bands that require a transceiver that can operate with robust performance over a wide range of frequencies, while maintaining low power consumption, low occupied volume, good noise performance, and low cost [1].

Low-noise amplifier (LNA) as the first active block of the receiver system determines the receiver bandwidth and noise figure (NF) [2]. Trade-offs between input matching, NF, gain, bandwidth, and linearity should be considered in the design process. Single-ended LNAs lack sufficient power supply rejection and have limited second-order distortion performance. Therefore, differential signaling is preferable because of its robustness against power supply, substrate noise, and second-order distortion [3]. Single-to-differential conversion (balun) is required to convert a single-ended RF signal into a differential signal. However, off-chip passive baluns are typically lossy and narrowband; thus, several baluns are needed to accommodate wideband operation, which increases the overall cost. A well-known topology for broadband applications employs a common-gate (CG) common-source (CS) pair as an active balun. This topology provides high power gain and the ability to cancel thermal noise and distortion over wideband frequency ranges [3,4].

Previous studies on CG-CS topology [3–12] show that a balanced condition between the CG and CS stages is essential for the noise canceling condition. Furthermore, scaling between the transconductance of the CS stage ($gm_{CS}$) and the transconductance of the CG stage ($gm_{CG}$) also affects the NF performance [3]. To satisfy input matching, $gm_{CG}$ must

be 20 $mS$. Thus, the drawback of this topology is its large power consumption, because $gm_{CS}$ needs to be significantly large to achieve better NF performance. Another critical issue is that the parasitic capacitances cause gain and phase imbalances due to the active circuit components. Previous work in [4] utilizes negative feedback from the source node of the cascoded CS stage to the gate of the CG stage. The voltage gain of the cascode CS stage is utilized to boost the $gm_{CG}$. The work in [5] improved the overall power consumption by employing negative feedback from the output of the CS stage to the gate of the CG stage, resulting in a higher effective $gm_{CG}$, reduced power consumption, and smaller size. Gain and phase imbalances can also be compensated using a passive capacitor between differential outputs. This technique is suitable for low-voltage applications, but its drawback is the headroom of the amplifier. Furthermore, the CS stage consumes a large amount of power because $gm_{CS}$ needs to be high for better noise performance. Other critical works in [6] and [8] proposed a balun CG-CS paired with the current reuse to further increase $gm/id$ efficiency but the trade-off between gain and bandwidth showed a marginal improvement factor. Authors in [12] proposed a CG-CS pair with the current reuse to decrease the power consumption and to improve the noise contribution of the CS stage as well, however the proposed design is just for a single-ended LNA and is not suitable for single to differential conversion.

This paper proposes a balun-LNA with CG-CS topology employing negative feedback on the CG stage and current reuse for the CS stage. Negative feedback utilizes the gain of the CS stage to increase $gm_{CG}$ for better input matching and to satisfy wideband operation. Current reuse is applied to increase the overall $gm_{CS}$ by adding a complementary transistor within the CS stage, resulting in higher transconductance scaling for better NF performance. Both negative feedback and current reuse keep the power consumption for the overall stages of the CG-CS topology low.

The remainder of this paper is organized as follows. Section 2 reviews the CG-CS active-balun topologies and their general properties. The analysis and properties of the current reuse technique are described in Section 3. The proposed LNA is described in Section 4, followed by a thorough analysis of its input matching, gain, and noise. Section 5 presents the simulation results of the proposed LNA, and conclusions are presented in Section 6.

## 2. CG-CS Active Balun Topologies

The circuit shown in Figure 1, proposed by [3], performs the balun operation over a wide frequency band and provides thermal noise cancelation of the CG stage, reducing the overall NF of the LNA. The input impedance of this topology is

$$Z_{in} = \frac{1}{gm_{CG}} \cdot \left(1 + \frac{R_{CG}}{rds_{CG}}\right), \tag{1}$$

where $gm_{CG}$, $rds_{CG}$, and $R_{CG}$ are the transconductance of $M_{CG}$, output resistance of $M_{CG}$, and load resistance of the CG stage, respectively. The input impedance of the LNA is approximately $(gm_{CG})^{-1}$ if $rds_{CG}$ is sufficiently large. Then, the input matching condition with an antenna's impedance (typically 50 $\Omega$) requires $gm_{CG} \approx 20\ mS$; this limits the CG stage voltage gain and necessitates a relatively high bias current. The voltage gain of the LNA is

$$Av = gm_{CG} \cdot Ro_{CG} + gm_{CS} \cdot Ro_{CS}, \tag{2}$$

where $Ro_{CG} = R_{CG} \parallel rds_{CG}$ and $Ro_{CS} = R_{CS} \parallel rds_{CS}$, and $gm_{CS}$, $Ro_{CS}$, and $rds_{CS}$ are the transconductance, resistance, and output resistance of the CS stage, respectively. The noise-canceling condition requires an equal gain for both stages with opposite phases. In addition, the scaling factor $n$ between the two stages ($gm_{CS} = n \cdot gm_{CG}$ & $R_{CS} = R_{CG}/n$) reduces the noise contribution of the CS stage. Increasing $n$ improves the NF of the LNA at the cost of increased power consumption. Perfect (balanced) conditions for noise-canceling are difficult to achieve because of the parasitic and process variations of passive devices.

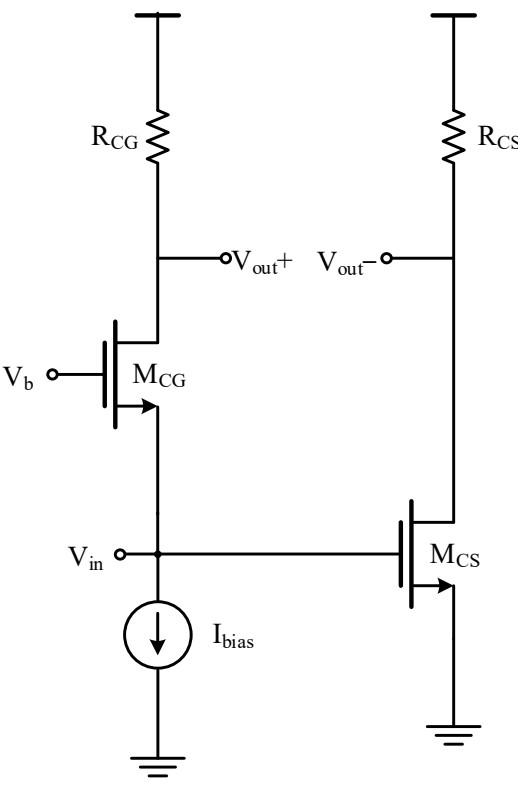

**Figure 1.** CG-CS active-balun LNA.

### 3. Current Reuse Technique

The current reuse technique was applied to our proposed circuit to increase the overall transconductance without increasing the total power consumption. This technique is suitable for the CG-CS topology because a high $gm_{CS}$ is necessary for better performance. In this section, we analyze the transconductance, linearity, and output resistance of the proposed scheme. Figure 2 shows an inverter-type current reuse circuit as a general circuit topology.

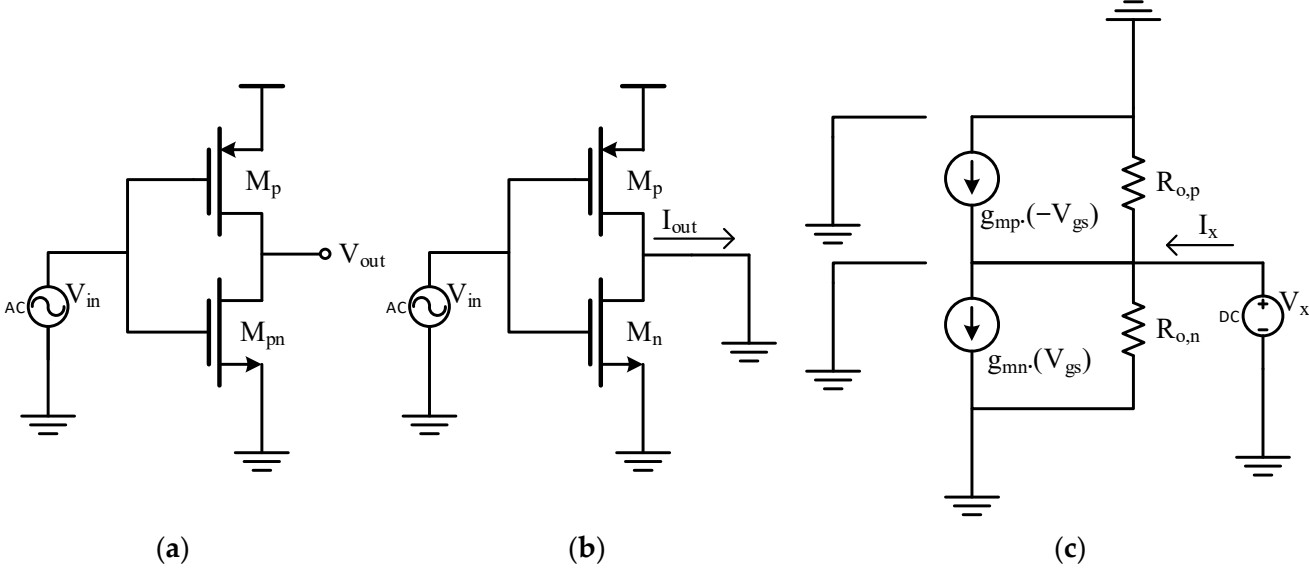

**Figure 2.** (**a**) Inverter-type current reuse circuit. (**b**) Scheme for transconductance calculation. (**c**) Equivalent small signal model for output resistance calculation.

### 3.1. Gm Calculation and Effect on Linearity

To calculate the transconductance ($G_m$), the output of the circuit is connected to the ground, as shown in Figure 2b. The total current sinked to the output ($I_{out}$) is

$$I_{out} = I_{o,p} - I_{o,n},\qquad(3)$$

where $I_{o,p}$ and $I_{o,n}$ are the output currents of the PMOS and NMOS transistors, respectively. The overall transconductance $G_m$ is expressed as

$$G_m = \frac{\partial\left(I_{o,p} - I_{o,n}\right)}{\partial Vin} = gm_p + gm_n.\qquad(4)$$

From Equation (4), $G_m$ is the sum of both the PMOS ($gm_p$) and NMOS ($gm_n$) transconductances. If both transistors are matched well, even-order distortions are canceled, leaving only the first-order (linear) term and higher odd-order components. The output current up to the third-order distortion terms is expressed as follows:

$$I_{out} = gm_p(-vin) + gm_{2,p}(-vin)^2 + gm_{3,p}(-vin)^3 - \left(gm_n(vin) + gm_{2,p}(vin)^2 + gm_{3,p}(vin)^3\right),\qquad(5)$$

$$I_{out} = -(gm_p + gm_n)(vin) + \left(gm_{2,p} - gm_{2,n}\right)(vin)^2 - \left(gm_{3,p} + gm_{3,n}\right)(-vin)^3,\qquad(6)$$

$$I_{out} \approx -(gm_p + gm_n)(vin) - \left(gm_{3,p} + gm_{3,n}\right)(-vin)^3,\qquad(7)$$

where $gm_{2,p/n}$, and $gm_{3,p/n}$ are second- and third-order nonlinear terms from the PMOS and NMOS transistors, respectively. Even though the third-order distortion component increased in Equation (7), the relative distortion with respect to the fundamental component is the same, and the third-order distortion component itself can be minimized by properly biasing both transistors.

### 3.2. Total Output Resistance

Figure 2c shows an equivalent small-signal model of the current reuse circuit. The drain-source resistance for both transistors is expressed by $Ro_{,p}$ and $Ro_{,n}$. To obtain the total output resistance, we null out any available input sources while injecting the test voltage source ($V_x$) into the output. Note that both inputs are grounded; thus, neither transconductance draws any current. Thus, the total current due to the test voltage ($I_x$) is

$$I_x = \frac{V_x}{R_{o,p}} + \frac{V_x}{R_{o,n}} = V_x\left(\frac{1}{R_{o,p}} + \frac{1}{R_{o,n}}\right),\qquad(8)$$

$$R_{out} = \left(Ro_{,p} \,//\, R_{o,n}\right),\qquad(9)$$

where $R_{out}$ is the total output resistance. This technique eliminates the usage of a resistor for the CS load in the CG-CS topology. Hence, the contribution of the resistor noise of the stage is reduced.

### 4. Wideband Balun-LNA with Feedback and Current Reuse Technique

Figure 3 shows the proposed wideband balun-LNA with the negative feedback for the CG stage and the current reuse technique for the CS stage. The CG stage provides input-matching and non-inverting signals at the CG output, whereas in the CS stage, the signal phase is inverted by 180°. The output of the CS stage is fed back to the gate of the CG stage to increase the effective $gm_{CG}$ by the gain of the CS stage. Thus, the negative feedback connection relaxes the input matching condition without increasing the current of the CG stage or increasing the transistor size for a higher $gm_{CG}$. In contrast, the CS stage employs complementary CMOS transistors using current reuse. The gain of each stage is

$$Av_{CS} = (gm_{CS,n} + gm_{CS,p}) \cdot Ro_{CS},\qquad(10)$$

$$Av_{CG} = gm_{CG}(1 + Av_{CS}) \cdot R_{CG}, \tag{11}$$

$$Ro_{CS} = (gm_{CCS}rds_{CCS}rds_{CS,n}//rds_{CS,p}), \tag{12}$$

where $Av_{CS}$ and $Av_{CG}$ are the gains of the CS and CG stages, respectively. Note that in (12), the CS stage output resistance $Ro_{CS}$ is determined by the transconductance of the cascoded NMOS transistor ($gm_{CCS}$) and its output resistance ($rds_{CCS}, rds_{CS,n}$) in parallel with the output resistance of the PMOS transistor ($rds_{CS,p}$). $Ro_{CS}$ can be approximated as $rds_{CS,p}$ because the cascoded NMOS transistor produces a much larger resistance at the CS output stage. The PMOS and NMOS transistors share the same current; therefore, the overall $Gm_{CS}$ is effectively doubled.

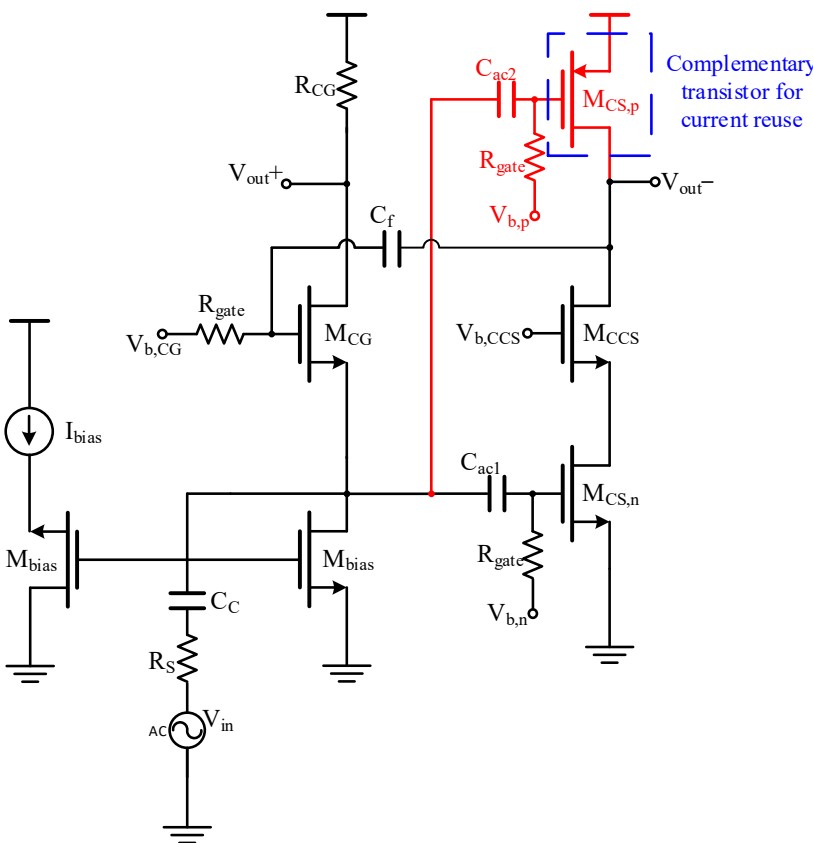

**Figure 3.** Proposed balun-LNA employing feedback and current reuse.

### 4.1. Input Matching

Note that in (11), the CG stage transconductance is increased by a factor of $1 + Av_{CS}$ due to the negative feedback employed from the CS stage output to the CG stage gate. Therefore, the input impedance of the proposed LNA can be approximated as

$$Z_{in} \cong \frac{1}{gm_{CG}(1 + Av_{CS})} // \frac{1}{sC_p}, \tag{13}$$

It is clear from Equation (13) that the required $gm_{CG}$ for the matching condition is reduced significantly, allowing the CG stage to have lower current consumption and size. The parasitic capacitance $C_p$, which is determined by the input pad, transistor size of the CG and CS stages, and bias transistors, is also reduced owing to the proposed scheme.

### 4.2. Noise Analysis

The important property of the CG-CS balun-LNA topology is that the thermal noise of the CG transistor is fully canceled when the gains of the CG and CS stages are balanced. The NF of the proposed LNA employing feedback and current reuse is

$$NF = 1 + \frac{\gamma gm_{CG} \cdot (R_{CG} - R_S \cdot gm_{CS} \cdot R_{CS})^2}{R_S \cdot Av_{diff}^2} + \frac{\gamma (gm_{CS,n} + gm_{CS,p}) \cdot Ro_{CS}^2 \cdot (1 + gm_{CG}(R_{CG} + R_S))^2}{R_S \cdot Av_{diff}^2} + \frac{R_{CG} \cdot (gm_{CG}(1 + Av_{CS}) \cdot R_S)^2}{R_S \cdot Av_{diff}^2}, \tag{14}$$

where $Av_{diff}$ is the differential gain, which can be expressed as

$$Av_{diff} = Av_{CG} + Av_{CS}. \tag{15}$$

The second term is the noise contribution owing to the CG stage, and its contribution is fully canceled when the balanced gain condition is met. The third term is the noise contribution due to the CS stage, and its noise is spread among the PMOS and NMOS transistors, whereas the last term is the noise due to the CG stage load resistance. Equation (14) does not consider the current mirror noise contributions, which can be significantly smaller with the feedback scheme [4].

## 5. Simulation Results

The balun-LNA was designed using TSMC 65 nm technology. Its device dimensions and bias points are indicated in Table 1. The balun-LNA layout is illustrated in Figure 4. The total area of the chip is $475 \times 366$ µm, including the balun-LNA core, common-mode feedback for gain and phase imbalance compensation, buffer (source follower) to interface balun-LNA outputs to off-chip ports, and pads. The active area of the LNA core is only $150 \times 150$ µm. The PMOS and NMOS CS stages are designed to match each other such that the second-order distortion produced by the stage is eliminated. The CG stage is designed to match a 50 Ω input impedance ($gm_{CG}(1 + Av_{CS}) = 20 \, mS$). As shown in Table 1, with a given transistor size, the CG stage only consumes 0.375 mA of current, in contrast with the conventional topology, which requires approximately 4 mA of current to obtain the same transconductance [2]. CS stage consumes the largest current in the overall stage by 3.7 mA. With the given size and bias condition, the transconductance of the CS stage is 2.5 times larger than the CG stage. The current reuse technique improved the $gm/id$ efficiency of the CS stage. Both positive output and negative output are sensed by common-mode feedback to ensure the stable gain and phase imbalance for robustness operation across PVT variation.

**Table 1.** Device dimension and bias point.

| Device | Dimension |
|---|---|
| $M_{CG}$ | 10/0.65 µm |
| $M_{CS,n}$, $M_{CCS}$ | 32/0.65 µm |
| $M_{CS,p}$ | 32/0.65 µm |
| $M_{Bias1}$ | 2/0.65 µm |
| $M_{Bias2}$ | 4/0.65 µm |
| $R_{CG}$ | 280 Ω |
| $R_{Gate}$ | 1.5 kΩ |
| $C_c$ | 10 pF |
| $C_{ac1}$, $C_{ac2}$ | 1.2 pF |
| $C_F$ | 5 pF |

Figure 5 shows a post-layout simulation of the input/output impedance matching performance ($|S_{11}|/|S_{22}|$) and power gain ($|S_{21}|$) of the total circuit and core, where the effect of the buffer is de-embedded. The simulated $|S_{11}|$ and $|S_{22}|$ values are well below $-10$ dB over a frequency range of 0.5–5 GHz. The maximum power gain of the proposed Balun-LNA is 20 dB and the maximum power gain with the effect of the buffer is 13 dB, including the intrinsic 6 dB loss from the output matching. The buffer was designed with a source follower architecture and consumes 4 mA of current, which can be avoided in the receiver front end, where the LNA is driving the mixer on-chip [1].

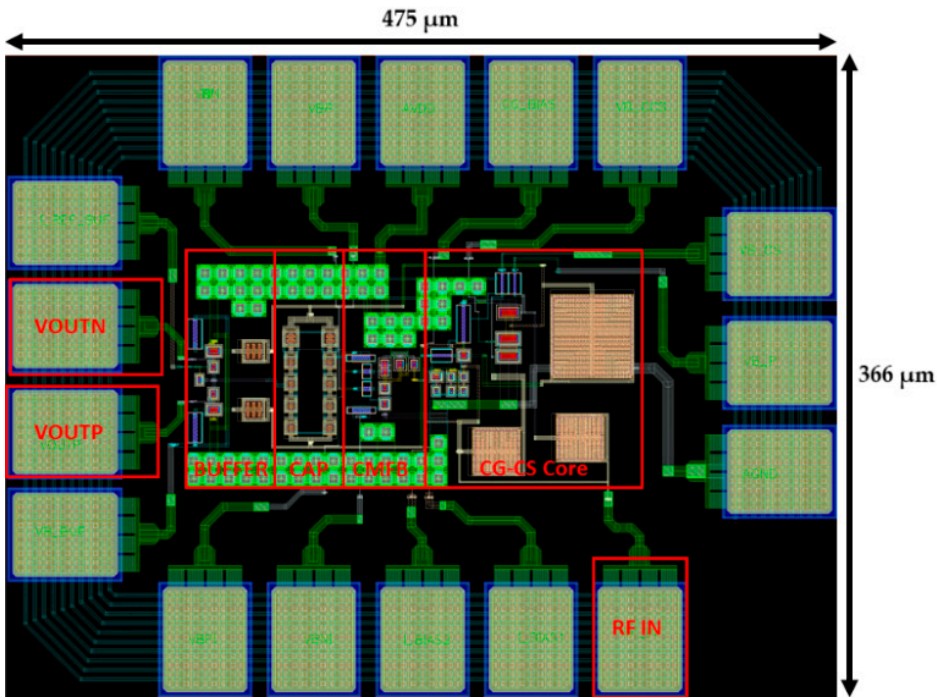

**Figure 4.** Layout of balun-LNA including core, feedback, and buffer.

Figure 6 shows the NF of the entire circuit (LNA core and buffer) and LNA core while de-embedding the buffer contribution, and the NF calculated from Equation (14) and designed device parameters is also shown. The calculated NF is 3.55 dB, while the LNA core shows an NF of 4 dB across the range of 0.5–2 GHz. The NF is increased in the post-layout simulation owing to the contribution of parasitic resistance, bias (current mirror), and other passive devices, such as $R_{Gate}$, which are not included in Equation (14). The CS stage provides the highest noise contribution for the LNA core, which is spread among the PMOS and NMOS transistors. With the buffer effect, the overall NF is increased by 0.8 dB owing to the contribution of the source follower and passive RC connection for high pass response. Within the frequency of operation, the NF for both the LNA core and the whole circuit shows favorable and stable results.

As mentioned in Equations (5)–(7), the current reuse technique affects third-order distortion performance. The distortion of the CG stage is also canceled along with the NF as long as both stages are in a balanced condition. IIP3 of the proposed LNA is mostly determined by the linearity of the CS stage itself. Linearity tests were performed using two different tones located at 2.4 GHz and 2.45 GHz (50 MHz spacing) (Figure 7a) and sweeping the injection tones from 0.5 GHz to 5 GHz with 50 MHz and 100 MHz tone spacing (Figure 7b). IIP3 of the LNA shows a minimum −12 dBm for both 50 MHz and 100 MHz tone spacing. Increased IIP3 across the frequency occurred due to lower gain at a higher frequency, which also reduces higher frequency harmonics.

Wideband LNA with noise-canceling ability has to offer stable performance against process, voltage, and temperature (PVT) variations. To ensure the robustness of the circuit, the core LNA is simulated across typical, slow-slow (SS), and fast-fast (FF) process corners at room temperature (27°) as well as different temperatures at −25°, and 80° Celsius as depicted in Figure 8. The power gain of the core LNA (Figure 8a,b) shows a maximum of 19 dB, 20 dB, and 20.7 dB for −25°, 27°, and 80° Celsius, respectively. This result shows an insensitive response of the proposed LNA against temperature variations. SS corner gives a maximum power gain of 18 dB, while FF corner gives 18.9 dB. SS corner affected the power gain at a higher frequency by 2 dB, while FF corner affected the gain at a lower frequency. However, both SS and FF corners are still showing high gain performance of the proposed LNA.

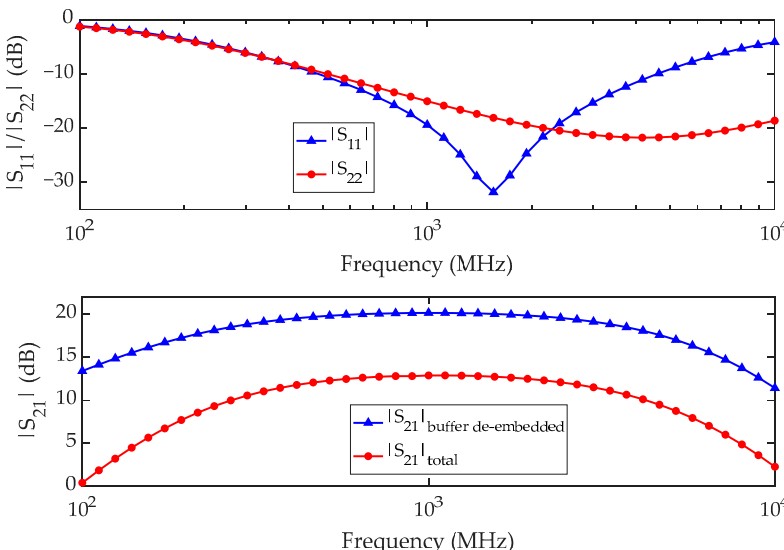

**Figure 5.** Simulation results for input/output matching ($|S_{11}|/|S_{22}|$) and power gain ($|S_{21}|$).

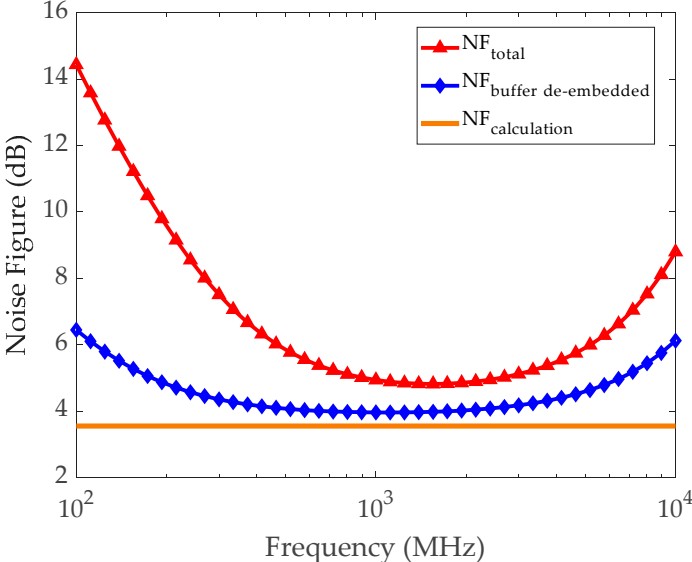

**Figure 6.** NF of proposed LNA versus frequency.

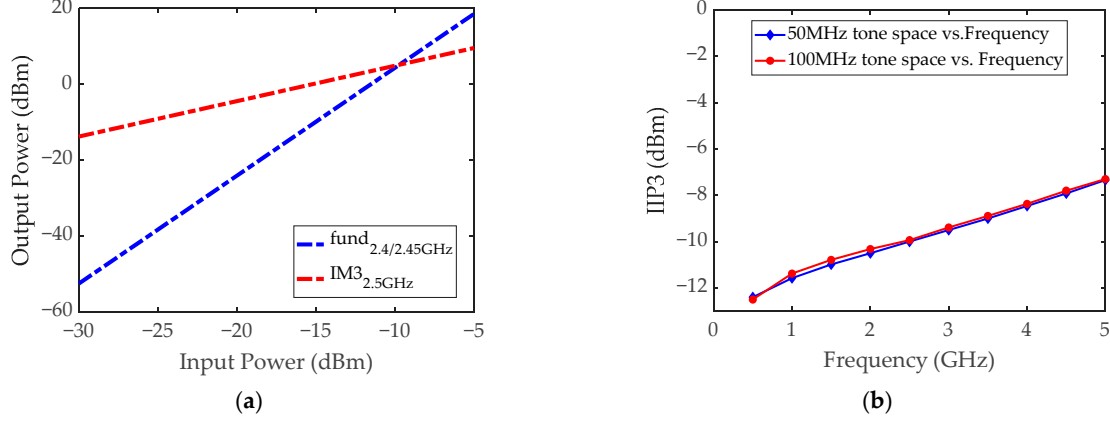

**Figure 7.** IIP3 of proposed LNA (**a**) at $fo = 2.4$ GHz with 50 MHz tone space (**b**) across 0.5 to 5 GHz frequency range with 50 and 100 MHz tone spacing.

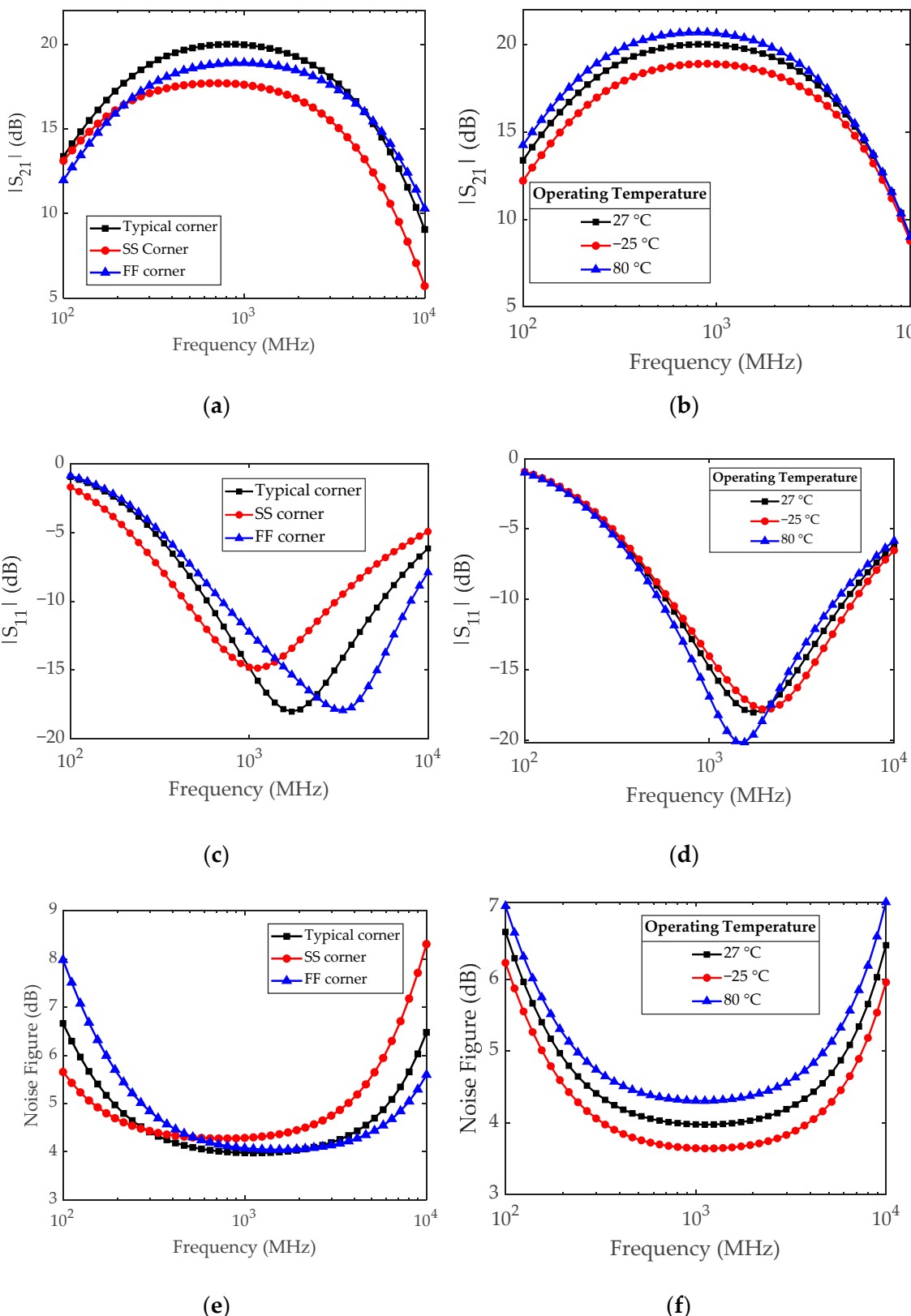

**Figure 8.** Performance of the proposed balun-LNA: (**a**) $|S_{21}|$ across corners, (**b**) $|S_{21}|$ across temperatures, (**c**) $|S_{11}|$ across corners, (**d**) $|S_{11}|$ across temperatures, (**e**) NF across corners, (**f**) NF across temperatures.

Input matching (Figure 8c,d) of the proposed LNA shows robustness against temperature variations. At −10 dB the point of low temperature shifted to 0.5–5.2 GHz, while high

temperature shifted to 0.45–4.8 GHz. Furthermore, process corners are shifting the −10 dB point of the LNA by 0.4–3.3 and 0.65–8 GHz for SS, and FF corners, respectively. The noise figure of an amplifier is mainly determined by its white (thermal) noise performance. Therefore, the NF corner simulation (Figure 8e,f) of the proposed shows temperature dependency. NF at −25°, 27°, and 80° are 3.65 dB, 4 dB, and 4.3 dB, respectively. Process variations at the SS and FF corners (Figure 8e) increase the noise contribution in higher frequency and lower frequency, respectively. The minimum NF of the SS corner is 4.25 dB, while the FF corner is 4 dB. All of $|S_{21}|$, $|S_{11}|$, and NF corners simulation results show small variation which confirms that our design is robust against PVT variation.

Furthermore, Monte Carlo (MC) simulations with 300 samples were run for $|S_{21}|$ and NF to evaluate the stability (robustness) of gain and minimum noise figure against the statistical variations as shown in Figure 9. Both simulations were carried out across 0.1–10 GHz frequency. Maximum $|S_{21}|$ (Figure 9a) gives a mean of 19.55 and a standard deviation of 1.925. Minimum NF (Figure 9b) of the proposed LNA gives a mean of 3.998 dB and a standard deviation of 0.182 dB. NF that is lower than 4.2 dB is 93% of total samples. Both maximum $|S_{21}|$ and minimum NF show stable results across the mismatch and process variability.

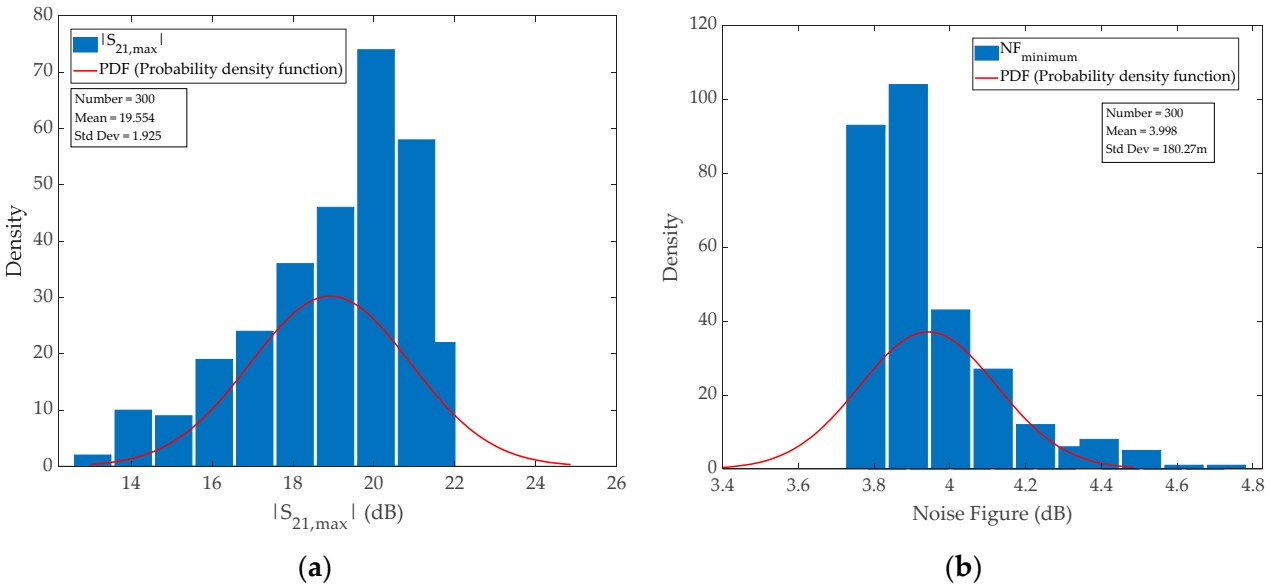

**Figure 9.** Monte Carlo simulation results across 0.1–10 GHz for: (**a**) maximum $|S_{21}|$, (**b**) minimum NF.

In [13], the phase imbalance was required to be within ±5° to suppress local oscillator (LO) leakage by more than 25 dBc. The gain and phase imbalance between the positive and negative outputs for the total circuit is shown in Figure 10. The parasitic from the devices and interconnect routings is compensated with the aid of common-mode feedback. The gain and phase imbalance of the LNA is only 0.15 to −0.54 dB and 0.2° to −0.3°, respectively, over the frequency of operation (0.5–5 GHz). Both results show effective and stable operation for the noise canceling condition as well.

The overall performance of the proposed LNA and previous works for balun-LNAs is compared in Table 2. Conventional CG-CS balun LNA topology [3] shows higher bandwidth and lower NF, but the power consumption is much higher than our design. Feedback CG-CS LNA without current reuse [5] has slightly lower power consumption, gain, and noise figure but exhibits much lower bandwidth than our work, which shows better power efficiency due to the proposed current reuse technique. The design proposed by [8] applied the same complementary NMOS/PMOS technique and has a high gain with low power consumption but gain and bandwidth are very limited due to the dependency of load resistance, compared to the bandwidth of our work which is significantly higher. LNA with active balun proposed by [10] has the highest gain among other designs but

at the cost of linearity degradation and increased power consumption. For performance comparison, the following figure of merit (FoM) [14,15] is used, which is typically used for LNA and low noise circuits.

$$FoM = \frac{G_{max}[Lin].BW(GHz)}{P_{DC}[mW].(NF_{min}[Lin]-1)},$$

where $G_{max}$, $BW$, $P_{DC}$, $NF_{min}$ are maximum gain, bandwidth, power consumption, and noise figure, respectively. Our LNA-balun shows the highest FOM among other works. Our proposed design further shows a fine performance for wideband operation, gain, and noise with minimum power consumption.

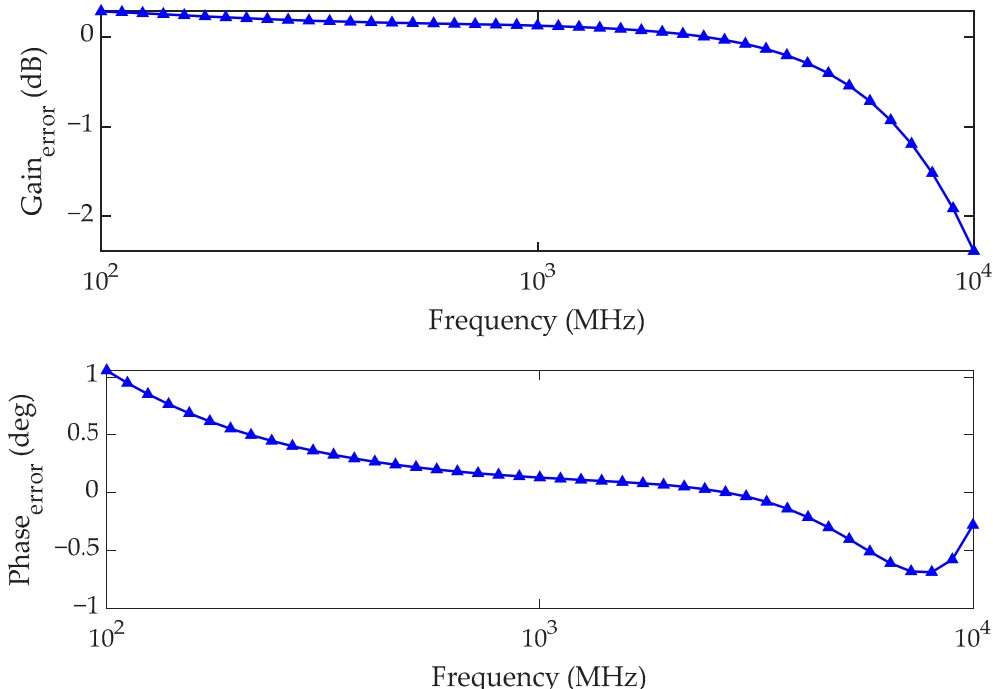

**Figure 10.** Gain and phase imbalance of proposed LNA.

**Table 2.** Performance comparison table.

| Parameter | This Work [S] | [3] [M] | [5] [M] | [6] [S] | [7] [M] | [8] [M] | [9] [M] | [10] [M] |
|---|---|---|---|---|---|---|---|---|
| Frequency [GHz] | 0.5–5 | 0.2–5.2 | 0.1–2 | 0.21–1.1 | 1.2–2 | 0.13–0.93 | 0.1–1 | 3–5 * |
| Gain [dB] | 20 | 15.6 | 16 | 24–30 | 16 | 16.6–19.6 | 14 | 30 * |
| Gain Imbalance [dB] | 0.15–(−0.54) | 0.7 | 0.5 | 2 | 0.6 | - | 1.4 | 0.4 |
| Phase Imbalance [deg] | 0.2–(−0.3) | 2 | 5 | 15 | 7 | - | - | 1.8 |
| NF [dB] | 4–4.5 | <3.5 | 3.8–5 | 2.8–3.8 | 3.8 | 3.6–5 | 4 | 3.6–4.3 * |
| IIP3 [dBm] | −10 | >0 | 0.5 | −13 | - | >−8.5 | 2 | −24 * |
| Power [mW] | 5 | 21 | 3 | 5.58 | 9.2 | 3 | 2.7 | 19 * |
| Supply [V] | 1.2 | 1.2 | 1.2 | 1.8 | 1.2 | 1.8 | 1.2 | 1.2 |
| Area [mm²] | 0.173 [a] | 0.01 [b] | 0.075 [b] | 0.24 [a] | - | 0.18 [b] | 0.0992 [b] | 0.734 [a] |
| Technology [CMOS] | 65 nm | 65 nm | 130 nm | 180 nm | 65 nm | 180 nm | 130 nm | 130 nm |
| FoM | 6 | 1.19 | 2.87 | 5.6 | 0.39 | 2.11 | 1.11 | 2.59 |

[S]: Simulated result; [M]: Measurement result; [a]: Total area; [b]: Active area; *: Measurement result including effects of buffer and active balun.

## 6. Conclusions

A wideband low-power balun LNA employing negative feedback and current reuse was presented in this paper. In the proposed scheme, the gain of the CS stage is further utilized by employing a negative feedback connection to increase the effective $gm_{CG}$, providing better input matching, while a complementary PMOS transistor is applied at the CS stage using the current reuse technique to improve the overall $gm_{CS}$ for better transconductance scaling. The current reuse technique can also eliminate the second-order distortion produced by the CS stage. These techniques reduce the overall power consumption, while the wideband frequency of operation, high gain performance, and NF are well maintained.

The proposed LNA is designed using the TSMC 65 nm technology. Compared with the other balun LNAs listed in Table 2, our design shows a high gain and high bandwidth, while keeping NF and power consumption low. Without any on-chip inductor, the total area occupied by the circuit is only $475 \times 366$ μm.

**Author Contributions:** Conceptualization and design, M.F.M. and J.K.; formal analysis, M.F.M.; writing—original draft preparation, M.F.M.; writing—review and editing, J.K.; supervision, J.K. and D.-H.L.; funding acquisition, J.K. All authors have read and agreed to the published version of the manuscript.

**Funding:** This research was jointly supported by the Regional Innovation Strategy (RIS) of the National Research Foundation of Korea (NRF) funded by the Ministry of Education (MOE) (2021RIS-004) and the Basic Science Research Program through the National Research Foundation of Korea (NRF) funded by the Ministry of Education (2021R1I1A304418211).

**Data Availability Statement:** Not applicable.

**Conflicts of Interest:** The authors declare no conflict of interest.

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
