# Peer review of "A Wideband Low-Power Balun-LNA with Feedback and Current Reuse Technique"

_electronics, doi:10.3390/electronics11091372_

Round 1
Reviewer 1 Report
The paper is generally well written but can be inproved. More discussion and comparison with the state of the art should be presented. In other word the actual table 2 should be better discussed showing not only performance improvements but also tehcnical novelty. References can also be improved, e.g. doi: 10.1109/INMMIC.2018.8430001
Author Response
Please see the attachment for our reponses.

Reviewer 2 Report
Electronics 2022
The paper by Muhammad Fakhri Mauludin, Dong-Ho Lee and Jusung Kim entitled “A Wideband Low-Power Balun-LNA with Feedback and Current
Reuse Technique” presents a low-noise amplifier (LNA) with single to differential conversion (Balun) for multi-standard radio applications. The proposed LNA combines a common-gate (CG) stage for wideband input matching and a common-source (CS) stage to cancel the noise and distortion of the CG stage. Using the proposed technique, a low noise figure (NF) is achieved while providing a wideband of operation. Furthermore, a feedback connection from the CS stage to the gate of the CG is employed to boost the transconductance of the CG stage (???? ), and an additional complementary transistor is applied at the CS stage using current reuse to increase the overall trans- conductance of the CS stage (????) without increasing the power consumed by the stage. This LNA was designed using TSMC 65 nm technology, and post-layout simulation results show operation across 0.5–5 GHz, a maximum power gain of 20 dB, 4 dB minimum NF, and third-order intercept point (IIP3) of -10 dBm while consuming only 5 mW of power from a 1.2 V supply.
The paper is organized as follows. After Section I reflecting an introduction to the problem, Section II reviews the CG-CS active-balun topologies and their general properties. The analysis and properties of the current reuse technique are described in Section III. The proposed LNA is described in Section IV, followed by a thorough analysis of its input matching, gain, and noise. Section V presents the simulation results of the proposed LNA, and conclusions are presented in Section VI.
Figure 3 illustrates proposed balun-LNA employing feedback and current reuse.
Input Matching, Noise Analysis are given in Section 4.
Simulation Results are described in Section 5.
The overall performance of the proposed LNA and previous works for balun-LNAs 227is compared in Table II. This table reflects that the proposed design shows a fine performance for wideband operation, gain,
and noise with minimum power consumption compared to other works.
Important: Use an unique numbering style in the whole manuscript, in present form two different numbering is used such as at the end of Introduction Section. It is written that “The remainder of this paper is organized as follows. Section II reviews the CG-CSactive-balun topologies and their general properties. The analysis and properties of thecurrent reuse technique are described in Section III. The proposed LNA is described in
Section IV, followed by a thorough analysis of its input matching, gain, and noise. SectionV presents the simulation results of the proposed LNA, and conclusions are presented in Section VI.”
That means roman type lettering for numbering. But the section captions are numbered using normal style such as
- Introduction, , 2. CG-CS Active Balun Topologies, 3. Current Reuse Technique, 4. Wideband Balun-LNA with Feedback and Current Reuse Technique, 5. Simulation Results, 6. Conclusion.
Improve the manuscript by using the same type of numbering!
I think that the paper reflects useful material and advances in the related field, with suggested minor improvement given above the paper can be considered for publication in the journal.
Reviewer 3 Report
The authors have presented a wideband low power LNA with feedback and current reuse technique. Although the proposed LNA is not fabricated and measured, the proposed circuit is analyzed and the obtained post-layout results are desirable, compared to the other related LNAs. The manuscript can be accepted after following modifications.
- Only 10 references are discussed in the introduction section. Kindly explain more recent works and describes more related approaches in the introduction section.
- In Figure 5 and related text it should be noted that the magnitude of S21/S11/S22 (|S21|/|S11|/|S22|) should be written, not S21/S11/S22.
- In Table 1, the second column (Dimension (W/L) μm) and third column (Bias mA) are not suitable for Resistors can capacitors.
- Figure 7 (IIP3) needs more explanations in the text. Also, kindly add linearity plot versus frequency in the manuscript.
Reviewer 4 Report
Dear Authors,
Your paper presents the design of a wideband LNA for multi-standard radio.
Overall, the paper is well writen and welle organized. All the statements are clearly justified. Even if there is no real novemlty in the LNA architecture, the paper is a good paper for people intended to gain knowledge in the domain.
Nevertheless, I have a concern with the size of the transistors which are all of miminum lenght. Whithout any corners and most importantly mismatch simulation, it is difficult to have a good idea of the design robustness. You may have to consider adding some monte carlos simulations of the design.
Second point, is there any FOM for LNA that could be used to fairly compare the different state of the art implementations?
Less important point, usually it is asked to place figure before any reference to it.
Reviewer 5 Report
Good piece of IC design craft for LNA. No layout / technology analysis reported - corner analysis is a must. ALWAYS. Nominal analysis neglect tech-node related effects which can degrade parameters expected (bandwidth, SNR...)
Round 2
Reviewer 1 Report
The paper can be accepted
Reviewer 3 Report
The authors have addressed all of my comments in the manuscript. The paper can be accepted in the present form.